# Effects of Renaming Schizophrenia on Destigmatization among Medical Students in One Taiwan University

**DOI:** 10.3390/ijerph18179347

**Published:** 2021-09-04

**Authors:** Yi-Hang Chiu, Meei-Ying Kao, Kah Kheng Goh, Cheng-Yu Lu, Mong-Liang Lu

**Affiliations:** 1Department of Psychiatry and Psychiatric Research Center, Wan Fang Hospital, Taipei Medical University, Taipei 116, Taiwan; chiuyihang@gmail.com (Y.-H.C.); havicson@gmail.com (K.K.G.); 2Graduate Institute of Humanities in Medicine, Taipei Medical University, Taipei 110, Taiwan; mykao@tmu.edu.tw; 3Department of Psychiatry, School of Medicine, College of Medicine, Taipei Medical University, Taipei 110, Taiwan; 4Psychology of Mental Health Programme, School of Health in Social Science, College of Arts, Humanities and Social Sciences, University of Edinburgh, Edinburgh EH8 9AG, UK; yaya880427@gmail.com

**Keywords:** renaming, public stigma, self-stigma, social distance, schizophrenia, medical students

## Abstract

The stigma associated with serious mental illnesses causes a huge burden on patients, their families, and society. In October 2012, in Taiwan, schizophrenia was renamed to reduce the stigma associated with this disease. The aim of this study was to compare the differences of public stigma, self-stigma, and social distance associated with schizophrenia between old and new name of schizophrenia in medical students. A cross-sectional survey was administered to 180 medical students of Taipei Medical University from October 2014 to February 2015. In total, 123 complete questionnaires were included in this study. Participants completed the modified attribution questionnaire, the perceived psychiatric stigma scale, and modified social distance scale to assess public stigma, self-stigma, and social distance, respectively. We also collected basic demographic data and previous experience of contact with people with mental illness. In total, 52 and 71 of the first- and fourth-year medical students, respectively, participated in the study. Among them, there were 51 females and 72 males. A significant difference in age was observed between the first- and fourth-year groups (20.2 ± 1.7 years vs. 22.7 ± 0.9 years, *p* < 0.001). After renaming schizophrenia, we noted significant differences in the scores in the modified attribution questionnaire, the perceived psychiatric stigma scale, and the modified social distance scale in all participants and the fourth-year students, respectively. Female gender (Beta = 0.230, *p* = 0.018) was significantly associated with the difference in the score of the modified attribution questionnaire after name change. The difference in the score of the perceived psychiatric stigma scale after the name change (Beta = 0.277, *p* = 0.004) and age (Beta = −0.186, *p* = 0.049) were significantly associated with the difference in the score of the modified social distance scale after name change. In conclusion, renaming was associated with the changes in the scores of the modified attribution questionnaire, the perceived psychiatric stigma scale, and the modified social distance scale toward individuals with schizophrenia in medical students of one Taiwan university. Further studies with large sample sizes, diverse participant backgrounds, and that monitor the subsequent behavioral changes are warranted.

## 1. Introduction

Originating in ancient Greece, the word “stigma” is referred to a symbol that is deliberately marked on an individual whose actions or moral standing has been deemed unacceptable by society. Stigmatization happens when a person deviates from social norms, and such stigma cannot be removed easily. Stigmatized individuals tend to experience social isolation and gradually conform to the stereotype associated with the stigma. Stigmatization affects the victim’s mental health, self-esteem, and behavior. It can harm their identity and engagement with society [1,2,3].

Several studies have shown that people with mental disorders avoid seeking help because they are wary of the harm from being stigmatized [4,5]. Numerous efforts have been made worldwide to reduce the stigma associated with mental illness. Several interventions, including simulations, media campaigns, education programs, and social exchange programs, have been developed to reduce mental illness related stigma and increase awareness [6]. Most interventions are based on the knowledge–attitudes–behavior model, in which the changes in human behavior are divided into three successive processes: the acquisition of knowledge, the generation of attitudes, and the formation of behavior [7]. Specifically, mental illness related stigma is proposed to be related to people’s lack of professional knowledge, their negative attitudes, and their desire for social distance from individual perceived as having a mental illness [8]. Negative attitudes by the majority of people toward individuals with mental illness give rise to stigma and can lead to discriminatory behavior [9]. A recent meta-analysis reported that interventions focusing on social contact or education have immediate, small-to-medium effects on the reduction in stigma toward people with severe mental illness [10].

In recent years, different names have been used to refer to schizophrenia in a country’s given language [11]. In Taiwan, the Chinese term for schizophrenia was “mind-splitting disease,” implying danger, uncertainty, severity, and nonrecovery. Thus, in October 2012, the Taiwanese Society of Psychiatry introduced a new name for schizophrenia that meant “disorder with dysfunction in thought and perception” [12]. The new name reflects the two major clinical characteristics of schizophrenia, dysfunction of thought and perception, and the word “disorder” suggests its treatability [12].

In 2002, the Japanese Society of Psychiatry and Neurology renamed the Japanese translation of schizophrenia from “seishin-bunretsu-byo” (mind-split disease) to “togo-shitcho-sho” (integration disorder) [13]. Study reported that the old Japanese name for schizophrenia is strongly associated with the Japanese term for “criminal,” whereas this association in the new name is remarkably weaker [14]. Studies conducted in Japan reported that the name change has reduced the stigma toward schizophrenia in short and long terms [15,16]. In 2012, the Korean Neuropsychiatric Association changed the original Korean name for schizophrenia, “jeongshin-bunyeol-byung” (mind-split disorder), to “johyun-byung” (attunement disorder) [17]. One study reported that the renaming has influenced the attitude toward schizophrenia in doctors and people from diverse backgrounds, such as mental health practitioners, patients, guardians, and university students in Korea [18]. A recent review article revealed that renaming schizophrenia is associated with improvements in attitudes toward patients with schizophrenia and increased diagnosis announcement [19].

Recently, a text-mining analysis of newspaper articles has shown no remarkable differences in the use of negative words between articles containing old and new names for schizophrenia in Taiwan [20]. To the best of our knowledge, the population study for investigating the effect of renaming schizophrenia on stigma in Taiwan is lacking in the literature.

The theoretical framework of stigma can be the tripartite model, which proposes that stigma is a combination of problems related to knowledge (ignorance), attitude (prejudice), and behavior (discrimination) [21]. The tripartite model is adaptable, allowing for attitudes towards individuals with mental illness to comprise the different dimensions of stigma [22]. These dimensions of stigma include public stigma, self-stigma, and social distance. Additionally, these dimensions can be seen as important indicators of destigmatizing attitudes [23].

Apart from the stigma in the general population, several studies showed that stigmatizing attitudes toward people with mental illness among health professionals and medical students exist in higher proportions than those expected given the current knowledge on this topic [24,25]. More education in psychiatry and contact with patients with mental illness can reduce the reduce the stigma toward those with mental illness in medical students [26].

Most medical students will become frontline healthcare professionals. They may encounter patients with mental illness, and any prejudice may influence the quality of their medical service. In this study, we intended to investigate the attitude toward schizophrenia held by medical students before and after schizophrenia was renamed. We evaluated the effect of renaming schizophrenia in three dimensions of stigma, namely public stigma, self-stigma, and social distance. This study had the following objectives: (1) to evaluate the effect of renaming schizophrenia on stigma reduction among medical students; (2) to evaluate the effect of years in medical education on destigmatization; and (3) to investigate the potential influential factors (e.g., demographic characteristics, previous contact experience, and different stigma dimensions) on the reduction in stigma.

## 2. Methods

### 2.1. Participants

The survey was carried out on medical students in the School of Medicine of the Taipei Medical University, Taiwan, who attended their first year and their fourth year of classes from October 2014 to February 2015. After the introduction of the purpose and process of our study at a class meeting, students were invited to participate in our study voluntarily. During introduction, we illustrated the background information of renaming schizophrenia in Taiwan. The participants were aware that the two names of schizophrenia denoted the same condition. Our study was approved by the Institutional Review Board of Taipei Medical University (approval number: 201312002). The Institutional Review Board of Taipei Medical University waived the requirement for the investigators to obtain written consent forms for all participants. Confidentiality and privacy of the participants were ensured using an anonymous questionnaire. In total, 180 questionnaires were distributed of which 125 questionnaires were returned. After excluding incomplete questionnaires, 123 questionnaires were included in the analysis.

In Taiwan, medical education takes seven years in medical school. Psychiatry education is scheduled for the fourth year, consisting of classroom lectures on clinical characteristics of mental disorders, their biopsychosocial treatments, and mental healthcare organization, as well as tutorial clinical workshops and attendance of clinical facilities. Therefore, the fourth year students who participated in this survey had already received their education in psychiatry.

### 2.2. Assessment Tools

The students were assessed by filling out the self-administered questionnaires that comprised four sections that each inquired into basic information, public stigma, self-stigma, and social distance. We collected basic information, such as age, gender, year in college, religion, and previous experience of contact with people with mental illness. Regarding the previous experience of contact with people with mental illness, the contents of questionnaire contained three dichotomous questions. (“Do you ever have a relative with mental illness?” “Do you ever have a friend with mental illness?” and “Do you ever have a classmate with mental illness?”).

#### 2.2.1. Public Stigma

Public stigma generally means that people stereotype and prejudge individuals in a minor group (e.g., with schizophrenia). People with schizophrenia experience public stigma which might adversely influence all aspects of the lives, such as poverty, unemployment, homeless, lower rate of marriage, and social exclusion [5]. For public stigma assessment, we used a modified version of the Corrigan’s attribution questionnaire [27]. Due to the similarity after translation into Chinese, we extracted 20 items of the Corrigan’s attribution questionnaire according to experts’ opinions for this study. For example, the Chinese translations of “I would feel aggravated by Harry” and “How irritated would you feel by Harry” are similar. We decided to keep the first item and delete the second item. These 20 items were grouped into nine subscales, namely blame (e.g., I would think that it was Harry’s own fault that he is in the present condition); anger (e.g., I would feel aggravated by Harry); pity (e.g., I would feel pity for Harry); help (e.g., how likely is it that you would help Harry?); dangerousness (e.g., I would feel unsafe around Harry); fear (e.g., Harry would terrify me); avoidance (e.g., if I were an employer, I would interview Harry for a job); segregation (e.g., I think Harry poses a risk to his neighbors unless he is hospitalized); and coercion (e.g., if I were in charge of Harry’s treatment, I would require him to take his medication). When answering each item, the participants had to choose one number from 1 (absolutely not) to 9 (absolutely). The items in subscales pity, help, and avoidance were reversely scored. A composite measure of public stigma is derived by totaling the sum of all statements (range: 20–180). The higher the score, the more public stigma demonstrated. The internal consistency was good, as Cronbach’s α = 0.83 in old name version and Cronbach’s α = 0.82 in new name version. In agreement with previous study [28], an exploratory factor analysis of the modified attribution questionnaire in the old and new name of schizophrenia yielded a six-factor solution, accounting for 68.9 and 69.0% of the variance, respectively.

#### 2.2.2. Self-Stigma

Self-stigma generally means that people in a minor group (e.g., with schizophrenia) experience and internalize the stigma as well as suffer from remarkable impacts on self-esteem and self-efficacy [29,30]. We used the perceived psychiatric stigma scale [31] to assess self-stigma. Rather than paying attention to labor rights and equal treatment, which is more commonly done in the Western society, people in Taiwanese society focus more on the family and marriage. Therefore, the developers of the perceived psychiatric stigma scale incorporated the items related to family and marriage into the questionnaire. In this case, the perceived psychiatric stigma scale can be used to accurately assess self-stigma in Taiwan. To capture the participant’s subjective impressions, the questions were written in the first-person voice and the narrative fashion. Exploratory factor analysis of the 25 item perceived psychiatric stigma scale yielded a three-factor solution that accounted for 48.4% of the variance [31]. The first subscale was “social ostracism,” in which participants had to imagine themselves as patients with schizophrenia and imagine how they would be treated by others. Example items of social ostracism were, “If people know that I have schizophrenia, my interpersonal relationship will be damaged” or “If people know that I have schizophrenia, my reputation will be harmed.” The second subscale was “marital preclusion,” which asked the participant how much they agreed with various negative stereotypes about marrying someone with schizophrenia. Example items of marital preclusion were, “Because I have schizophrenia, I believe that nobody is willing to marry me” or “Because I have schizophrenia, I believe that the parents of my partner will oppose our marriage.” The third subscale was “self-deprecation,” which is defined as the condition of undervaluing oneself. Example items of self-deprecation were, “I am a weakling because I have schizophrenia” or “Because I have schizophrenia, I am ashamed of myself.” We used a four-point Likert scale with the categories of completely disagree, disagree, agree, and completely agree in this questionnaire. A composite measure of self-stigma is derived by totaling the sum of all statements (range: 25–100). The higher the score, the more self-stigma demonstrated. The Cronbach’s α was 0.94 and test–retest reliability with 1-week interval was 0.90 [31]. In accordance with the result of original research [31], exploratory factor analysis of the perceived psychiatric stigma scale in the old and new name of schizophrenia yielded a three-factor solution accounting for 48.5 and 47.6% of the variance, respectively.

To experience self-stigma, the person must be aware of the stereotypes and prejudice toward a stigmatized group (e.g., people with schizophrenia) as well as internalize them and suffer from numerous negative consequences [32]. Because most of the participants have never experienced mental health problems or contacted with people with mental illness, we asked participants to put themselves in a patient’s position and answer the perceived psychiatric stigma scale.

#### 2.2.3. Social Distance

A modified version of Bogardus’s social distance scale [33] was used to assess social distance. The measure of social distance is to assess a respondent’s willingness to interact with a described individual in different kinds of situations [34]. For example, individuals who believe that people with schizophrenia are dangerous are more likely to increase social distance from them [35]. Differences in social distance toward schizophrenia were observed between ethnic groups, genders, knowledge of schizophrenia, social classes, careers, religions, and countries [36]. We chose seven questions from the original scale that were best adapted to the daily lives of students in this study. Different distances represented different levels of intimacy. For example, the response, “I am willing to stay in the same city with the patient” indicated the farthest distance, and “I am willing to get married with a patient” indicated the closest distance. We used a Guttman scale with the categories of disagree and agree in this questionnaire. The Guttman scale is a cumulative scale designed so that agreement with higher-level responses assumes agreement with all lower-level responses. A total score ranges from 0 to 7, with higher scores reflecting higher levels of social distance.

### 2.3. Statistical Analysis

All statistical analyses were performed using IBM SPSS Statistics for Windows, version 19.0 (IBM Corp.). Descriptive statistics (mean ± standard deviation, frequency, and percentage) were calculated to illustrate the demographic characteristics of participants and study variables. Shapiro–Wilk’s method was used for normality test. Continuous variables were compared between groups using Student’s *t* test, and categorical variables were compared using the chi-square test or Fisher’s exact test. The false discovery rate (FDR) was applied for multiple testing corrections [37]. The differences between the groups were considered significant if *p* < 0.05.

The dependent variables were the differences in the scale scores between the old and new name of schizophrenia. The independent variable selection (e.g., demographic characteristics, previous contact experiences, and differences in other stigma scale scores) in the multiple linear regression model was based on the results of exploratory univariate linear regressions. Only variables that were significantly associated with dependent variables in the univariate analyses (with *p* < 0.05) were included as independent variables in a subsequent multiple linear regression.

## 3. Results

Among 123 respondents, 72 and 51 were men and women, respectively (Table 1). No differences were observed in all basic characteristics between the first- and fourth-year groups, except for age (20.2 ± 1.7 years vs. 22.7 ± 0.9 years, *p* < 0.001). The prevalence of previous experience of contact people with mental illness, including relatives, friends, and classmates, was not significantly different between two groups.

Significant differences were observed in all participants in the scores of the modified attribution questionnaire, the perceived psychiatric stigma scale, and the modified social distance scale after renaming schizophrenia (Table 2). Furthermore, for all participants, compared with the new name, the old name was associated with higher scores in the three subscales of the perceived psychiatric stigma scale, namely social ostracism, marital preclusion, and self-deprecation.

In the first-year medical students, significant differences were observed in the scores of the perceived psychiatric stigma scale, all three subscales of the perceived psychiatric stigma scale, and the modified social distance scale after the name change. In the fourth-year medical students, significant differences were observed in the scores of the modified attribution questionnaire, the perceived psychiatric stigma scale, social ostracism subscale, marital preclusion subscale, and the modified social distance scale after the name change.

To identify which variables were significantly associated with stigma outcomes, a multiple regression analysis was performed using the difference in stigma scale score between the old name and new name as the dependent variable, and various measures exhibited a significant correlation in a univariate linear regression analysis as independent variables. The difference in the score of the modified attribution questionnaire was correlated with female gender (Beta = 0.200, *p* = 0.038), the difference in the score of the modified social distance scale (Beta = 0.196, *p* = 0.042), and the difference in the score of the perceived psychiatric stigma scale (Beta = 0.195, *p* = 0.045) in univariate linear regressions. When we examined these factors together in multivariate linear regression, only female gender (Beta = 0.230, *p* = 0.018) remained significant associated with the difference in the score of the modified attribution questionnaire (Table 3).

The difference in the score of the perceived psychiatric stigma scale was correlated with the difference in the score of the modified social distance scale (Beta = 0.343, *p* < 0.001) and the difference in the score of the modified attribution questionnaire (Beta = 0.195, *p* = 0.045) in univariate linear regressions. When we examined these factors together in multivariate linear regression, only the difference in the score of the modified social distance scale (Beta = 0.302, *p* = 0.002) retained a significant association with the difference in the score of the perceived psychiatric stigma scale.

The difference in the score of the modified social distance scale was correlated with the difference in the score of the perceived psychiatric stigma scale (Beta = 0.343, *p* < 0.001), age (Beta = −0.201, *p* = 0.021), and the difference in the score of the modified attribution questionnaire (Beta = 0.196, *p* = 0.042) in univariate linear regressions. When we examined these factors together in multivariate linear regression, the difference in the score of the perceived psychiatric stigma scale (Beta = 0.277, *p* = 0.004) and age (Beta = −0.186, *p* = 0.049) retained a significant association with the difference in the score of the modified social distance scale (Table 4).

## 4. Discussion

To the best of our knowledge, this is the first population study to investigate the effect of renaming on the attitude toward schizophrenia in Taiwan. The results of our study showed that renaming schizophrenia produced significant differences in the scores of the modified attribution questionnaire, the perceived psychiatric stigma scale, and the modified social distance scale toward individuals with schizophrenia in medical students.

For public stigma, the score of the modified attribution questionnaire of the new name was significantly lower than that of the old name in all participants. This finding is consistent with that of a recently published review, which reported that name changes are associated with improvements in attitudes toward people with schizophrenia [19]. The multiple regression analysis showed that gender was considerably associated with the difference in the score of the modified attribution questionnaire. This finding is consistent with those of previous studies reporting that gender affects the change in public stigma toward psychosis [38,39].

With regard to self-stigma, the score of the perceived psychiatric stigma scale was significantly different between the new name and old name of schizophrenia in all participants. Furthermore, the scores of all three subscales of the perceived psychiatric stigma scale, namely social ostracism, marital preclusion, and self-deprecation, significantly differed after renaming schizophrenia. Among these decreases, that of marital preclusion was the most noteworthy. In traditional and even contemporary Taiwanese society, an ideal marriage should be between two families of equal social status. Many people are unwilling to accept people with schizophrenia into their families. Patients with schizophrenia thus feel a sense of inferiority and assume that their partner’s parents will reject them. A study pointed out that marital preclusion is a characteristic that cannot be ignored in Taiwanese society and is a consequence of the stigmatization of mental illness [31]. Renaming schizophrenia can reduce self-stigma, such as marital preclusion, implying its considerable impact and importance.

Our result showed that changing the name of schizophrenia changed the score of the modified social distance scale in all participants. This result is consonant with that for public stigma. People tend to distance themselves from individuals they stereotype as frightening, such as patients with schizophrenia [35,40]. Participants were less fearful of people with schizophrenia when the new name was used. A multiple regression analysis showed that age was significantly associated with the change in social distance. This finding is consistent with those of previous studies, which reported that age influenced the social distance individuals kept from people with psychosis [24,41].

Renaming schizophrenia significantly differed the score of the modified attribution questionnaire in the fourth-year medical students but not their first-year counterparts. Previous studies reported that being in professional contact with people with mental illness and having more knowledge of psychiatry help medical students view them in a more positive way [42,43]. A recent network meta-analysis showed that contact-based education is the most effective anti-stigma intervention in healthcare professionals and students [44]. In our study, the fourth-year medical students who had more knowledge in psychiatry and greater contact with patients with schizophrenia are probably a contributory factor of the result.

In the fourth-year medical students, the subscale score of self-deprecation did not decrease after schizophrenia was renamed. The fourth-year medical students had already received a clinical training course in psychiatry and had clinical contact with patients with schizophrenia in acute exacerbation. Some patients with schizophrenia might have functional impairment and low self-esteem. This clinical experience might have influenced the scoring of the self-deprecation subscale. A meta-analysis showed that contact with inpatients might lead to be pessimistic about recovery and has a small effect on stigma toward mental illness among medical students [45]. Based on our findings, more appropriate programs that provide contacts, not only to inpatients with acute exacerbation but also recovered outpatients in the community, would be beneficial and necessary. Therefore, clerkships in psychiatry that take place in a combination of inpatient and outpatient settings are more effective in changing medical students’ attitudes, as opposed to those settled in purely hospitalized conditions [45].

## 5. Limitations and Future Directions

There are some limitations on generalization of our study results. First, the study participants were medical students of Taipei Medical University, and selection bias was possible. Second, this study was a cross-sectional survey that could not determine any causal associations. Third, the findings of the current study in medical students may not reflect the actual attitudes of the lay public. Fourth, the small sample size may limit the generalization of our study. Fifth, the modified Corrigan’s attribution questionnaire and the modified Bogardus’s social distance scale were not fully validated in Taiwan. In addition, culture is important in determining which characteristics are stigmatized in different groups [46]. Adaptations of existing Western-developed stigma measures to Taiwanese warrant further investigations. Sixth, the majority of participants had no personal experience of mental illness. Even though they were asked to put themselves in patient’s position and fill out the questionnaire, the application of the perceived psychiatric stigma scale to assess self-stigma in people without schizophrenia made the result doubtful. Further studies to investigate the effect of renaming on self-stigma in people with schizophrenia are warranted. Seventh, the differences in scale scores, as proxy measures of destigmatization, might not guarantee their behavioral changes [16,47]. Eighth, the use of self-reported measures cannot be exempt from social desirability bias. Finally, the lack of sample size calculation might affect the power of our study.

Further studies in this field are recommended. It is desirable that the limitations of our study are taken into consideration. First, studies with large sample sizes and diverse backgrounds are warranted. Second, validation studies of existing stigma measures and culture-specific stigma measures should be performed. Third, prospective studies with multiple follow-up evaluation sessions are warranted to monitor the changes in attitude and behavior. Fourth, the development of appropriate psychiatry education for medical students to against stigma toward people with mental illness is important.

## 6. Conclusions

The results are in agreement with the general and clinical belief that renaming might reduce the stigma toward schizophrenia. We discovered that the scores of social distance, public stigma, and self-stigma were significantly different between new name and old name of schizophrenia among medical students. Further studies with large sample sizes, participants with diverse backgrounds, and that monitor the subsequent behavioral changes are warranted.

## Figures and Tables

**Table 1 ijerph-18-09347-t001:** Demographic characteristics of participants.

Characteristics	Total	First Year	Fourth Year	*p* Value ^a^
Gender				0.266
Female	51	25	26	
Male	72	27	45	
Age (year)	21.7 ± 1.8	20.2 ± 1.7	22.7 ± 0.9	<0.001 *
Religion				0.488
No	82	35	47	
Buddhism/Taoism	22	11	11	
Christian	19	6	13	
Relative with mental illness				0.834
No	93	40	53	
Yes	30	12	18	
Friend with mental illness				0.153
No	91	42	49	
Yes	32	10	22	
Classmate with mental illness				0.239
No	84	39	45	
Yes	39	13	26	

^a^ Comparison between the first- and fourth-year groups; * remained significant after FDR correction.

**Table 2 ijerph-18-09347-t002:** The effects of renaming schizophrenia on destigmatization.

	Total			First Year			Fourth Year		
	Old Name	New Name	*p* Value	Old Name	New Name	*p* Value	Old Name	New Name	*p* Value
Public stigma	101.5 ± 18.9	97.1 ± 18.9	<0.001 *	101.3 ± 21.0	99.1 ± 21.1	0.149	101.7 ± 17.5	95.7 ± 14.9	<0.001 *
Self-stigma	60.9 ± 12.3	58.2 ± 11.3	<0.001 *	59.1 ± 12.0	55.9 ± 11.1	<0.001 *	61.9 ± 12.5	59.8 ± 11.1	<0.001 *
Social ostracism	24.9 ± 5.9	23.8 ± 5.5	<0.001 *	24.1 ± 5.4	22.8 ± 5.3	0.003 *	25.4 ± 6.2	24.5 ± 5.5	0.003 *
Marital preclusion	22.0 ± 4.2	20.9 ± 4.5	<0.001 *	21.3 ± 4.7	19.9 ± 4.5	<0.001 *	22.5 ± 4.2	21.6 ± 3.9	<0.001 *
Self-deprecation	13.9 ± 3.4	13.4 ± 3.0	0.002 *	13.8 ± 3.2	13.0 ± 2.7	0.010 *	14.0 ± 3.5	13.7 ± 3.2	0.074
Social distance	4.0 ± 2.0	3.4 ± 2.0	<0.001 *	4.2 ± 1.9	3.4 ± 2.0	<0.001 *	3.8 ± 2.1	3.4 ± 2.0	0.003 *

* remained significant after FDR correction.

**Table 3 ijerph-18-09347-t003:** Multiple regression analysis with the difference in the score of the modified attribution questionnaire as the dependent variable.

Variables	Beta	t	*p* Value
Female gender	0.230	2.413	0.018
Difference in the score of modified social distance scale	0.188	1.878	0.063
Difference in the score of the perceived psychiatric stigma scale	0.158	1.600	0.113

**Table 4 ijerph-18-09347-t004:** Multiple regression analysis with the difference in the score of modified social distance scale as the dependent variable.

Variables	Beta	t	*p* Value
Difference in the score of the perceived psychiatric stigma scale	0.277	2.929	0.004
Age	−0.186	−1.993	0.049
Difference in the score of the modified public stigma scale	0.145	1.535	0.128

## Data Availability

The datasets generated for this study are available on request to the corresponding author.

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
