# Peer review of "Effects of Renaming Schizophrenia on Destigmatization among Medical Students in One Taiwan University"

_ijerph, 2021, doi:10.3390/ijerph18179347_

Round 1
Reviewer 1 Report
This work is interesting, today is a topical topic and requires important actions from the medical and social fields
I make some recommendations to improve your study
On line 64 you cite Takahashi et al. Reported.. remove the name and cite according to magazine rules. Do the same with the quote (Cho et al. On line 70)
In his study, there is an important and prominent variable that justifies the possible changes in perception of the disease. This variable is training (knowledge), fourth-year students have better results. Knowing the disease seems to improve people’s expectations of the disease. It should establish greater control over this variable so that it can really be justified whether it is the name change of the disease or the formation and knowledge of the disease that produces the improvement. At the heart, there is talk of an attitude towards the disease and as you know attitudes have a high component of knowledge. Attitudes improve with knowledge, I recommend you to argue and develop this idea in your introduction to avoid interpretation biases. The changes are also justified from this argument
I recommend you argue the attitudinal dimension towards the disease in your theoretical framework. There are many studies in this respect
Include in the data analysis section the justification for the characteristics of the sample distribution and whether the criteria for the use of parametric or non-parametric tests were met.
The discussion presents the limitations of the study. Please add a section on limitations and foresight to your work and include this in that section, not in the discussion.
Review references and cite correctly according to journal rules

Author Response
Response to Reviewers’ Comments
Dear Editor and Reviewers,
Thank you for giving us the opportunity to submit a revised draft of my manuscript titled “Effects of renaming schizophrenia on destigmatization among medical students in one Taiwan university” to International Journal of Environmental Research and Public Health. We appreciate the time and effort that you and the reviewers have dedicated to providing your valuable feedback on our manuscript. We are grateful to the reviewers for their insightful comments on our paper. We have been able to incorporate changes to reflect most of the suggestions provided by the reviewers. We have highlighted the changes within the manuscript. Here is a point-by-point response to the reviewers’ comments and concerns
Comments from Reviewer 1
This work is interesting, today is a topical topic and requires important actions from the medical and social fields
I make some recommendations to improve your study
Comment 1: On line 64 you cite Takahashi et al. Reported.. remove the name and cite according to magazine rules. Do the same with the quote (Cho et al. On line 70)
Response: Thanks for your comment. We have revised those sentences accordingly.
Comment 2: In his study, there is an important and prominent variable that justifies the possible changes in perception of the disease. This variable is training (knowledge), fourth-year students have better results. Knowing the disease seems to improve people’s expectations of the disease. It should establish greater control over this variable so that it can really be justified whether it is the name change of the disease or the formation and knowledge of the disease that produces the improvement. At the heart, there is talk of an attitude towards the disease and as you know attitudes have a high component of knowledge. Attitudes improve with knowledge, I recommend you to argue and develop this idea in your introduction to avoid interpretation biases. The changes are also justified from this argument
Response: Thanks for your suggestion. We have revised the second paragraph of the introduction. The following sentences have been added. “Most interventions are based on the knowledge-attitudes-behavior model which the changes of human behavior are divided into three successive processes: the acquisition of knowledge, the generation of attitudes, and the formation of behavior. Specifically, mental illness related stigma is proposed to be related to people’s lack of professional knowledge, their negative attitudes, and their desire for social distance from individual perceived as having a mental illness. Negative attitudes by the majority of people toward individuals with mental illness give rise to stigma and can lead to discriminatory behavior.”
Comment 3: I recommend you argue the attitudinal dimension towards the disease in your theoretical framework. There are many studies in this respect.
Response: Thanks for your suggestion. As response to previous comment, we have added the description about the knowledge-attitudes-behavior model. And we also have added a new paragraph to illustrate the theoretical framework of stigma. “The theoretical framework of stigma can be the tripartite model, which proposes that stigma is a combination of problems related to knowledge (ignorance), attitude (prejudice), and behavior (discrimination). The tripartite model is adaptable which it allows for attitudes towards individuals with mental illness to be comprised of the different dimensions of stigma. These dimensions of stigma include public stigma, self-stigma, and social distance. And these dimensions can be seen as important indicators of destigmatizing attitudes.”
Comment 4: Include in the data analysis section the justification for the characteristics of the sample distribution and whether the criteria for the use of parametric or non-parametric tests were met.
Response: Thanks for your comment. We have applied the Shapiro-Wilk's method for normality test. And we used the parametric tests for analysis.
Comment 5: The discussion presents the limitations of the study. Please add a section on limitations and foresight to your work and include this in that section, not in the discussion.
Response: Thanks for your comment. We have added the Section 5 Limitations and Future directions. “There are some limitations on generalization of our study results. First, the study participants were medical students of Taipei Medical University, and selection bias was possible. Second, this study was a cross-sectional survey that could not determine any causal associations. Third, the findings of the current study in medical students may not reflect the actual attitudes of the lay public. Fourth, small sample size may limit the generalization of our study. Fifth, the modified Corrigan’s attribution questionnaire and the modified Bogardus’s social distance scale were not fully validated in Taiwan. In addition, culture is important to determine which characteristics are stigmatized in different groups. Adaptations of existing Western-developed stigma measures to Taiwanese warrant further investigations. Sixth, the majority of participants had no personal experience of mental illness. Even they were asked to put themselves in patient’s position and fill out the questionnaire, the application of the perceived psychiatric stigma scale to assess self-stigma in people without schizophrenia made the result doubtful. Further studies to investigate the effect of renaming on self-stigma in people with schizophrenia are war-ranted. Seventh, the differences in scale scores, as proxy measures of destigmatization, might not guarantee their behavioral changes. Eighth, the use of self-reported measures cannot be exempt from social desirability bias. Finally, the lack of sample size calculation might affect the power of our study.
Further studies in this field are recommended. It is desirable that the limitations of our study are taken into consideration. First, studies with large sample sizes and diverse backgrounds are warranted. Second, validation studies of existing stigma measures and culture-specific stigma measures should be performed. Third, prospective studies with multiple follow-up evaluation sessions are warranted to monitor the changes in attitude and behavior. Fourth, the development of appropriate psychiatry education for medical students to against stigma toward people with mental illness is important.”
Comment 6: Review references and cite correctly according to journal rules
Response: Thanks for your comment. We have reviewed the references and cited correctly.
Comment 7: Extensive editing of English language and style required.
Response: The manuscript has been edited by Professor Winston W. Shen. Professor Shen is the Editor-in-Chief of Taiwanese Journal of Psychiatry and a Tenured Professor of Psychiatry at Saint Louis University.

Reviewer 2 Report
In this paper the authors used data collected from 123 medical students in a medical school of Taiwan to examine the relation between public stigma, self -stigma and social distance and the use of the term schizophrenia and a new term to reduce stigma in this group. The paper deals with an interesting and important issue. However, my concern is about methodological procedures, specifically the factorial validity of the scales. This analysis needs further substantial work before publication. Below are a few comments:
Minor concerns
- The title should specify that the study was made in only one university in medical students.
From the abstract, considering the limitations of the study, I think the authors could be modest with their claims.
- The introduction of the paper is mostly well written. However, the stigma and discrimination concepts might be further distinguished to avoid confusion. On line 165:” The higher the score the more discrimination and stigmatization demonstrated”.
- I think it is better to write gender than sex.
Major concens
Method.
- The "measures" sub-section needs re-writing and editing.
- Data on questionnaires require more clarity. Is there a cultural adaptation in Taiwan? How the items were selected by the experts? Please include some examples.
- All the information about the factorial validity of the measures is missing. The authors should include factor structure of the subscales of the modified Attribution Questionnare (e.g. blame, anger, help, fear….) and Perceived Psychiatry Stigma Scale (e.g. social ostracism, mental preclusion, and self-deprecation).
Discussion.
- It is important to include cultural aspects (See: Yang, L, et.al., (2007) adding culture and stigma), because a huge part of the research was made in occidental countries. The authors could include the implications of these findings in Taiwan and mention some contrast with other countries .
Yang, L, Thornicroft, G., Alvarado, R., et al. (2014). Recent advances in cross-cultural measurement in psychiatric epidemiology: utilizing ‘what matters most’ to identify culture-specific aspects of stigma, International Journal of Epidemiology, Volume 43, Issue 2, April 2014, Pages 494–510, https://doi.org/10.1093/ije/dyu039
- What is needed to continue with this line of research?
Other limitations should include as potential generalization of the results found, the use of self-report questionnaires, or the lack of sample size calculation.
References: There are many references, include the most relevant for the study.
This reference could be useful to improve the discussion:
Yin-Yi Lien, Hui-Shin Lin, Yin-Ju Lien, Chi-Hsuan Tsai, Ting-Ting Wu, Hua Li & Yu-Kang Tu (2021) Challenging mental illness stigma in healthcare professionals and students: a systematic review and network meta-analysis, Psychology & Health, 36:6, 669-684, DOI: 10.1080/08870446.2020.1828413
Author Response
Dear Editor and Reviewers,
Thank you for giving us the opportunity to submit a revised draft of my manuscript titled “Effects of renaming schizophrenia on destigmatization among medical students in one Taiwan university” to International Journal of Environmental Research and Public Health. We appreciate the time and effort that you and the reviewers have dedicated to providing your valuable feedback on our manuscript. We are grateful to the reviewers for their insightful comments on our paper. We have been able to incorporate changes to reflect most of the suggestions provided by the reviewers. We have highlighted the changes within the manuscript. Here is a point-by-point response to the reviewers’ comments and concerns
Comments from Reviewer 2
Reviewer 2
In this paper the authors used data collected from 123 medical students in a medical school of Taiwan to examine the relation between public stigma, self -stigma and social distance and the use of the term schizophrenia and a new term to reduce stigma in this group. The paper deals with an interesting and important issue. However, my concern is about methodological procedures, specifically the factorial validity of the scales. This analysis needs further substantial work before publication. Below are a few comments:
Minor concerns
Comment 1: The title should specify that the study was made in only one university in medical students.
Response: Thanks for your comment. We have revised the title. “Effects of renaming schizophrenia on destigmatization among medical students in one Taiwan university”
Comment 2: From the abstract, considering the limitations of the study, I think the authors could be modest with their claims.
Response: Thanks for your comment. We have revised the abstract accordingly. “In conclusion, renaming was associated with the changes in the scores of the modified attribution questionnaire, the perceived psychiatric stigma scale, and the modified social distance scale toward individuals with schizophrenia in medical students of one Taiwan university.”
Comment 3: The introduction of the paper is mostly well written. However, the stigma and discrimination concepts might be further distinguished to avoid confusion. On line 145:” The higher the score the more discrimination and stigmatization demonstrated”.
Response: Thanks for your comment. We have revised the sentence as the following. “The higher the score, the more public stigma demonstrated.”
Comment 4: I think it is better to write gender than sex.
Response: Thanks for your suggestion. We have revised it accordingly.
Major concerns
Method.
Comment 5: The "measures" sub-section needs re-writing and editing.
Response: The manuscript has been edited by Professor Winston W. Shen. Professor Shen is the Editor-in-Chief of Taiwanese Journal of Psychiatry and a Tenured Professor of Psychiatry at Saint Louis University.
Comment 6: Data on questionnaires require more clarity. Is there a cultural adaptation in Taiwan?
Response: Thanks for your comment. There is no cultural adaptation in the modified attribution questionnaire and the modified social distance scale. But the perceived psychiatric stigma scale was developed by Taiwanese psychologists. As mentioned in original version of our paper, rather than paying attention to labor rights and equal treatment in the Western society, people in Taiwanese society focus more on the family. We have re-written the section 2.2.2. “Rather than paying attention to labor rights and equal treatment, which is more commonly done in the Western society, people in Taiwanese society focus more on the family and marriage. Therefore, the developers of the perceived psychiatric stigma scale incorporated the items related to family and marriage into the questionnaire. In this case, the perceived psychiatric stigma scale can be used to accurately assess self-stigma in Taiwan.”
Comment 7: How the items were selected by the experts? Please include some examples.
Response: Thanks for your comment. We have added the following sentences to illustrate the process of item selection. “For example, the Chinese translations of “I would feel aggravated by Harry” and “How irritated would you feel by Harry” are similar. Then we decided to keep the first item and delete the second item.”
Comment 8: All the information about the factorial validity of the measures is missing. The authors should include factor structure of the subscales of the modified Attribution Questionnare (e.g. blame, anger, help, fear….) and Perceived Psychiatry Stigma Scale (e.g. social ostracism, mental preclusion, and self-deprecation).
Response: Thanks for your comment. We have added the following sentences in the section 2.2.1 to illustrate the factor structure of the modified attribution questionnaire. “In agreement with previous study, exploratory factor analysis of the modified attribution questionnaire in old and new name of schizophrenia yielded a six-factor solution ac-counting for 68.9% and 69.0% of the variance, respectively.”
And we have added the following sentences to present the factor structure of the perceived psychiatry stigma scale. “In accordance with the result of original research, exploratory factor analysis of the modified attribution questionnaire in old and new name of schizophrenia yielded a three-factor solution accounting for 48.5% and 47.6% of the variance, respectively.”
Discussion.
Comment 9: It is important to include cultural aspects (See: Yang, L, et.al., (2007) adding culture and stigma), because a huge part of the research was made in occidental countries. The authors could include the implications of these findings in Taiwan and mention some contrast with other countries .
Yang, L, Thornicroft, G., Alvarado, R., et al. (2014). Recent advances in cross-cultural measurement in psychiatric epidemiology: utilizing ‘what matters most’ to identify culture-specific aspects of stigma, International Journal of Epidemiology, Volume 43, Issue 2, April 2014, Pages 494–510
Response: Thanks for your suggestion. We have revised the limitations accordingly. “Fifth, the modified Corrigan’s attribution questionnaire and the revised Bogardus’s social distance scale were not fully validated in Taiwan. In addition, culture is important to determine which characteristics are stigmatized in different groups [47]. Adaptations of existing Western-developed stigma measures to Taiwanese warrant further investigations.”
And we have mentioned the culture differences between Taiwan and other countries in section 2.2.2. “Rather than paying attention to labor rights and equal treatment, which is more commonly done in the Western society, people in Taiwanese society focus more on the family and marriage. Therefore, the developers of the perceived psychiatric stigma scale incorporated the items related to family and marriage into the questionnaire. In this case, the perceived psychiatric stigma scale can be used to accurately assess self-stigma in Taiwan.”
We also have mentioned the culture differences between Taiwan and other countries in the third paragraph of discussion. “In traditional and even contemporary Taiwanese society, an ideal marriage should be between two families of equal social status. Many people are unwilling to accept people with schizophrenia into their families. Patients with schizophrenia thus feel a sense of inferiority and assume that their partner’s parents may reject them. A study pointed out that marital preclusion is a characteristic that cannot be ignored in Taiwanese society and is a consequence of the stigmatization of mental illness.”
Comment 10: What is needed to continue with this line of research?
Response: Thanks for your suggestion. We have added the following sentences in Limitations and Future Directions section. “Further studies in this field are recommended. It is desirable that the limitations of our study are taken into consideration. First, studies with large sample sizes and diverse backgrounds are warranted. Second, validation studies of existing stigma measures and culture-specific stigma measures should be performed. Third, prospective studies with multiple follow-up evaluation sessions are warranted to monitor the changes in attitude and behavior. Fourth, the development of appropriate psychiatry education for medical students to against stigma toward people with mental illness is important.”
Comment 11: Other limitations should include as potential generalization of the results found, the use of self-report questionnaires, or the lack of sample size calculation.
Response: Thanks for your comment. We have revised the limitations section. “There are some potential limitations on generalization of our study results. First, the study participants were medical students of Taipei Medical University, and selection bias was possible. Second, this study was a cross-sectional survey that could not determine any causal associations. Third, the findings of the current study in medical students may not reflect the actual attitudes of the lay public. Fourth, small sample size may limit the generalization of our study. Fifth, the modified Corrigan’s attribution questionnaire and the modified Bogardus’s social distance scale were not fully validated in Taiwan. In addition, culture is important to determine which characteristics are stigmatized in different groups. Adaptations of existing Western-developed stigma measures to Taiwanese warrant further investigations. Sixth, the majority of participants had no personal experience of mental illness. Even they were asked to put themselves in patient’s position and fill out the questionnaire, the application of the perceived psychiatric stigma scale to assess self-stigma in people without schizophrenia made the result doubtful. Further studies to investigate the effect of renaming on self-stigma in people with schizophrenia are warranted. Seventh, the differences in scale scores, as proxy measures of destigmatization, might not guarantee their behavioral changes. Eighth, the use of self-reported measures cannot be exempt from social desirability bias. Finally, the lack of sample size calculation might affect the power of our study.”
Comment 12: References: There are many references, include the most relevant for the study.
Response: Thanks for your comment. We have reviewed the references and decreased the number of references.
Comment 13: This reference could be useful to improve the discussion:
Yin-Yi Lien, Hui-Shin Lin, Yin-Ju Lien, Chi-Hsuan Tsai, Ting-Ting Wu, Hua Li & Yu-Kang Tu (2021) Challenging mental illness stigma in healthcare professionals and students: a systematic review and network meta-analysis, Psychology & Health, 36:6, 669-684, DOI: 10.1080/08870446.2020.1828413
Response: Thanks for your recommendation. We have cited the reference and revised the last two paragraphs in the discussion section. “Renaming schizophrenia significantly differed the score of the modified attribution questionnaire in the fourth-year medical students but not their first-year counterparts. Previous studies reported that being professionally involved with people with mental illness and more knowledge in psychiatry help medical students view them in a more positive way. A recent network meta-analysis showed that contact-based education is the most effective anti-stigma intervention in healthcare professionals and students. In our study, the fourth-year medical students who had more knowledge in psychiatry and greater contact with patients with schizophrenia might probably be a contributory factor of the result.
In the fourth-year medical students, the subscale score of self-deprecation did not de-crease after schizophrenia was renamed. The fourth-year medical students had already received a clinical training course in psychiatry and had clinical contact with patients with schizophrenia in acute exacerbation. Some patients with schizophrenia might have functional impairment and low self-esteem. This clinical experience might have influenced the scoring of the self-deprecation subscale. A meta-analysis showed that contact with inpatients might lead to be pessimistic about recovery and has a small effect on stigma toward mental illness among medical students. Based on our findings, more appropriate programs that provide contacts not only to inpatients with acute exacerbation but also recovered outpatients in the community would be beneficial and necessary. Therefore, clerkships in psychiatry that take place in a combination of inpatient and outpatient settings are more effective in changing medical students’ attitudes, as opposed to those settled in purely hospitalized conditions.”

Round 2
Reviewer 2 Report
The authors took into consideration the comments I wrote in the first review, the article improved significantly. I don’t have any additional comments